# Gender differences in predictors of intensive care units admission among COVID-19 patients: The results of the SARS-RAS study of the Italian Society of Hypertension

Guido Iaccarino[1]*, Guido Grassi[2], Claudio Borghi[3], Stefano Carugo[4], Francesco Fallo[5], Claudio Ferri[6], Cristina Giannattasio[7], Davide Grassi[6], Claudio Letizia[8], Costantino Mancusi[1], Pietro Minuz[9], Stefano Perlini[10], Giacomo Pucci[11], Damiano Rizzoni[12], Massimo Salvetti[13], Riccardo Sarzani[14], Leonardo Sechi[15], Franco Veglio[16], Massimo Volpe[17], Maria Lorenza Muiesan[13], on Behalf of the SARS-RAS Investigators[¶]

1 Dept. of Advanced Biomedical Sciences, Federico II University, Naples, Italy, 2 Department of Medicine and Surgery, University of Milano-Bicocca, Milano, Italy, 3 Dept. of Medicine and Surgery Sciences, Alma Mater Studiorum University of Bologna, Bologna, Italy, 4 Università degli Studi di Milano, Milano, Italy, 5 Clinica Medica 3, Department of Medicine (DIMED), University of Padova, Padova, Italy, 6 Dept. of Clinical Medicine, Public Health, Life and Environment Sciences, University of L'Aquila, L'Aquila, Italy, 7 Cardiology 4, "A. De Gasperis" Department, ASST GOM Niguarda Ca' Granda Hospital, Milan, Italy, 8 Department of Translational and Precision Medicine, Unit of Secondary Arterial Hypertension, Policlinico "Umberto I", "Sapienza" University of Rome, Rome, Italy, 9 Department of Medicine, Section of Internal Medicine C, University of Verona, Verona, Italy, 10 Department of Internal Medicine and Therapeutics, University of Pavia, Pavia, Italy, 11 Section of Internal Medicine Terni, Department of Medicine, University of Perugia, Terni, Italy, 12 Department of Clinical and Experimental Sciences, University of Brescia-Internal Medicine ASST Spedali Civili PO Montichiari, Brescia, Italy, 13 Dept. of Clinical & Experimental Sciences, Medicina 2, ASST Spedali Civili Brescia, University of Brescia, Brescia, Italy, 14 Internal Medicine and Geriatrics, Department of Clinical and Molecular Sciences, Italian National Research Centre on Aging, Hospital "U. Sestilli", IRCCS-INRCA, University "Politecnica Delle Marche", Ancona, Italy, 15 Department of Medicine, University of Udine, Udine, Italy, 16 Division of Internal Medicine and Hypertension, Department of Medical Sciences, University of Turin, Turin, Italy, 17 Clinical and Molecular Medicine Department, Rome and IRCCS Neuromed, Sapienza University Sant'Andrea Hospital, Pozzilli (IS), Italy

☯ These authors contributed equally to this work.
¶ Membership of SARS-RAS Investigator group is provided in the Acknowledgments.
* guiaccar@unina.it

**Data Availability Statement:** Data are available at DOI: 10.6084/m9.figshare.12622208.

## Abstract

### Background

The global rate of intensive care unit (ICU) admission during the COVID-19 pandemic varies within countries and is among the main challenges for health care systems worldwide. Conflicting results have been reported about the response to coronavirus infection and COVID-19 outcomes in men and women. Understanding predictors of intensive care unit admission might be of help for future planning and management of the disease.

### Methods and findings

We designed a cross-sectional observational multicenter nationwide survey in Italy to understand gender-related clinical predictors of ICU admission in patients with COVID-19. We

**Funding:** This work was supported by the Italian Society of Hypertension, who provided secretariat, logistic, communication, publication cost coverage. The funders had no additional role in study design, data collection and analysis, decision to publish, or preparation of the manuscript.

**Competing interests:** The authors have declared that no competing interests exist.

analyzed information from 2378 charts of Italian patients certified for COVID-19 admitted in 26 hospitals. Three hundred ninety-five patients (16.6%) required ICU admission due to COVID19 infection, more frequently men (74%), with a higher prevalence of comorbidities (1,78±0,06 vs 1,54±0,03 p<0.05). In multivariable regression model main predictors of admission to ICU are male gender (OR 1,74 95% CI 1,36–2,22 p<0.0001) and presence of obesity (OR 2,88 95% CI 2,03–4,07 p<0.0001), chronic kidney disease (OR: 1,588; 95%, 1,036–2,434 p<0,05) and hypertension (OR: 1,314; 95% 1,039–1,662; p<0,05). In gender specific analysis, obesity, chronic kidney disease and hypertension are associated with higher rate of admission to ICU among men, whereas in women, obesity (OR: 2,564; 95% CI 1,336–4.920 p<0,0001) and heart failure (OR: 1,775 95% CI: 1,030–3,057) are associated with higher rate of ICU admission.

## Conclusions

Our study demonstrates that gender is the primary determinant of the disease's severity among COVID-19. Obesity is the condition more often observed among those admitted to ICU within both genders.

## Trial registration

Clinicaltrials.gov: NCT04331574.

## Background

The global rate of intensive care unit (ICU) admission during the COVID-19 pandemic varies within countries and is among the main challenges for health care systems worldwide. Understanding predictors of ICU admission might be of help for future planning and management of the disease. The clinical response to coronavirus infection in men and women appears to be different according to scientific reports [1]. A gender-specific analysis has not yet been carried out, while most recently, researchers' attention focused on pregnant women infected by SARS-Cov2 infection [2].

In China, the gender distribution was equal in a group of 140 patients with COVID-19 [3], and in a subsequent report of 1099 patients with COVID-19 from 552 hospitals in 30 Chinese provinces, 58% of the patients were men [4]. Some more recent data from the New York City area show a lower prevalence of women, and confirm lower mortality for different age range [5]. Female gender seems to represent a protective factor for in-hospital mortality [odds ratio 0.79 (CI 0.65–0.95)] in an extensive observational database collecting patients from Asia, Europe, and the United States [6].

Among critically ill patients, fewer women were affected than men in China [7] and Italy [8] (33 and 18% respectively). The analysis of gender-disaggregated data has shown a similar number of men and women affected by the disease, although mortality seems to be higher in men. It has been suggested that the observed male gender predisposition could be explained by the higher proportion of smokers in men than in women [9]. In addition to lifestyle habits, other gender-related aspects, including enzymatic activity, metabolism and immunology, and drug response, have been evoked as potential explanations for these observations [10].

In the last few months, it has become clear that Age and multimorbidity are the significant determinants of more severe clinical manifestation of the disease [11–13]. Some authors also

hypothesize that ACE inhibitors and Angiotensin Receptor Antagonist (ARB) can induce protection from COVID-19 [14].

To the best of our knowledge, whether gender-related differences in clinical characteristics, comorbidities, and treatment may affect an adverse outcome is still not established. Nevertheless, such a piece of information is very much expected and called for [15]. Accordingly, we explored the influence of gender-related differences, comorbidities, and treatment on clinical predictors of ICU admission in patients with COVID-19.

## Methods and findings

### Study population

The SARS-RAS is a cross-sectional, multicenter, observational study conducted in 26 hospitals and centers in Italy [13]. The centers were distributed in 13 regions, each contributing according to the detailed geographical distribution of the disease, most of the patients being located in Northern regions, especially Lombardy, compared to Southern regions. The patients' cohort included 2378 patients aged 18 to 101 years with confirmed COVID-19, according to World Health Organization interim guidance [16]. The observation period started on March 9th and ended on April 29th, 2020. The study is performed under the article 89 of the General Data Protection and Regulation, which allows the processing of personal data for archiving purposes in the public interest, scientific or historical research purposes or statistical purposes, provided that technical and organizational measures are in place to ensure the principle of data minimization (https://gdpr-info.eu). The SARS-RAS study is registered in Clinicaltrials.gov at the accession number NCT04331574.

### Procedures

An online questionnaire was distributed among the centers to collect reviewed epidemiological, clinical, and outcomes data from hospital emergency rooms, regular and intensive care wards. Each center designated at least one physician that was instructed to the acquisition and review of the requested information. Patients were pseudonymized by assigning a deidentified identification code. The questionnaire collected information regarding the center and the Age, gender, nationality (Italian or other), and city of origin of the patient. From the anamnesis, we collected whether the patient had a known diagnosis of hypertension with prescribed antihypertensive drugs, coronary artery disease (history of myocardial Infarction, PCI or CABG), heart failure (based on clinical history), diabetes (with prescribed antidiabetic drugs), chronic kidney disease [based on anamnestic estimated glomerular filtration rate (eGFR) below 60 ml/min/kg], chronic obstructive pulmonary disease (based on the presence of signs and symptoms according to GOLD 2019), obesity (body mass index $\geq$30 Kg/m$^2$ according to the Center for Disease Control and Prevention; https://www.cdc.gov/obesity/adult/defining.html), history of blood and solid tumors, liver disease, and other comorbidities; pharmacological treatment as regards the use of RAS inhibitors (ACE inhibitors, ARBs) and other antihypertensives; the degree of the severity of COVID-19 [17]. The electronic data was transmitted with the modern cryptography systems over the web and stored in a locked, password-protected computer. All collected records were then quality checked by two authors (G.I and C.M.).

COVID-19 diagnosis was confirmed in all patients by RT-PCR performed on nasopharyngeal or throat swab samples [18] in each center by the designated Institutions.

We also collected the outcomes (hospital dismission or death) if available at the time point of the survey. All patients for which the course of the disease was in an active state, were classified as such [16].

## Statistical analysis

Descriptive analyses of the variables were expressed as mean and standard errors (S.E.) or frequencies expressed in absolute numbers and percentages. We used ANOVA to analyze continuous variables, and the $\chi^2$ test or the Kruskal Wallis test as appropriate to compare categorical data. We tested regression analyses, odds ratio, and confidence intervals on the interest variables grouped by admission or non-admission to the intensive care unit (ICU). We performed multivariable regression analyses on the significant and clinically relevant continuous and categorical variables to assess the primary determinant of ICU admission. We applied the same model after grouping the study population by gender. $p < 0.05$ was considered statistically significant.

## Results

Answers to all questions of the questionnaire were mandatory before the online form could be validated. We collected 2409 questionnaires. After a quality check, 2378 were used for the analysis. Reasons for discharge were incongruences between the responses or duplication. The average Age was 68,21±0,38 years, and 1489 (62.6%) were men; patients were prevalent of Italian nationality (94%). Patients presented with several comorbidities including hypertension, diabetes, chronic obstructive pulmonary disease, coronary artery disease, heart failure, obesity, chronic kidney disease; the prevalence of diabetes, coronary artery disease, and chronic kidney disease was lower in women than in men while, as expected, thyroid diseases were more prevalent in women (Table 1). Women took fewer ACE inhibitors and alpha-blockers and more diuretics than men (Table 2).

Of the total study population, 395 patients entered ICUs. These patients were more frequently men, with a higher prevalence of obesity, hypertension, diabetes, chronic kidney disease, and heart failure (all $p < 0.05$, Table 1).

The use of ACE inhibitors or ARBs medications did not differ among patients admitted or not admitted to ICU. Interestingly, ACE inhibitors were less used, while diuretics were more often prescribed in women admitted to ICUs (Table 2).

The rate of admission in ICU was higher for males, compared to females (19,5 vs 11.7%, $p < 0.0001$). Compared to men, women admitted to ICU were older, with a more considerable prevalence of heart failure. (Table 3, Fig 1, panel A and B).

**Table 1. Demographic characteristics of the study population.**

|  | Total study population (n = 2378) | Males (n = 1489) | Females (n = 889) | p | Admitted to ICU (n = 395) | Not admitted to ICU (n = 1983) | p |
|---|---|---|---|---|---|---|---|
| **Age (years)** | 68,2141±0,38 | 66,9±0,39 | 70,3±0,77 | *0,0005* | 68.9±0.70 | 68.1±0.43 | 0,418 |
| **Men (%)** | 62.6 |  |  |  | 73.7 | 60.4 | *0.0005* |
| **Hypertension (%)** | 58.5 | 58.8 | 58.0 | 0,689 | 65,3 | 57.2 | *0.003* |
| **Obesity (%)** | 6,6 | 6,7 | 6,5 | 0.861 | 12.4 | 5.5 | *0.0001* |
| **Diabetes (%)** | 18,2 | 19,8 | 15,5 | *0.009* | 22,8 | 17,3 | *0.01* |
| **COPD (%)** | 8,5 | 8,9 | 7,5 | 0.280 | 10.4 | 8.1 | 0.132 |
| **CKD (%)** | 6.1 | 6,4 | 4.3 | *0.031* | 8.6 | 5.0 | *0.004* |
| **Coronary artery disease (%)** | 14.3 | 16,9 | 10.0 | *0.0005* | 15,7 | 14,1 | 0.153 |
| **Heart Failure (%)** | 12.1 | 11,9 | 12,4 | 0.716 | 15.2 | 11,6 | 0.05 |
| **Thyroid Disease (%)** | 0,2 | 0,1 | 0,5 | *0,049* | 0,0 | 0,3 | 0,318 |

COPD: Chronic obstructive pulmonary disease; CKD: Chronic kidney disease; p-value for Age was calculated by unpaired *t*-test; for categorical variables, the $\chi^2$ test was used.

**Table 2. Cardiovascular active drugs in the study population.**

|  | Total study population (n = 2378) | Males (n = 1489) | Females (n = 889) | p | Admitted to ICU (n = 395) | Not admitted to ICU (n = 1983) | p |
|---|---|---|---|---|---|---|---|
| ACE-inhibitors (%) | 22.2 | 24,3 | 18,6 | **0,001** | 24,6 | 21.7 | 0.211 |
| ARBs (%) | 18.6 | 17,7 | 20,2 | 0,141 | 16,7 | 19.1 | 0.281 |
| β-blockers (%) | 23.6 | 24,2 | 22,5 | 0,358 | 24.3 | 23.4 | 0.703 |
| Ca-Antagonists (%) | 8,1 | 8,2 | 8,0 | 0,864 | 7,8 | 8.2 | 0.829 |
| Diuretics (%) | 15,4 | 14,2 | 17,4 | **0,042** | 14,7 | 15.5 | 0.667 |
| α-blockers | 1,9 | 2,9 | 0,2 | **0,001** | 1,8 | 1,9 | 0,847 |

ACE: Angiotensin-converting enzyme; ARB: Angiotensin II Receptor 1 Blockers; **β**-Adrenergic receptor blockers p values for categorical variables, were calculated by the $\chi^2$ test.

In the multivariable regression model, the main predictors of admission to ICU were male gender, obesity, hypertension, and chronic kidney disease (Table 4, Fig 2, panel A).

In the gender-specific analysis, among men, obesity, hypertension, and chronic kidney disease were associated with a higher rate of admission to ICU (Fig 2, panel B). Similarly, among women, obesity and heart failure were associated with a higher rate of ICU admission (Fig 2, panel C).

## Discussion

Admission to ICU has been a significant challenge for all health care systems, fighting the COVID-19 pandemic. The percentage of patients admitted to ICU differs among countries, ranging from 5 to 32% in China and 5 to 10% in Italy. Knowledge of clinical characteristics of patients admitted to ICU is of crucial importance for future management of the disease, with specific regards to the prevention and surveillance of active patients with mild to moderate disease. Our study demonstrates that the main determinants of ICU admission are male gender and obesity and that patients admitted to ICU have more comorbidities than those not-admitted to ICU. Furthermore, a gender-specific phenotype of patients admitted to ICU exists since men are more often obese, hypertensive, and affected by chronic kidney disease, while women are older than men and present with obesity and heart failure.

**Table 3. Characteristics of the ICU population.**

|  | Admitted to ICU (n = 395) | Males (n = 291) | Females (n = 104) | p |
|---|---|---|---|---|
| Age (years) | 68.9±0.70 | 67,4±0,81 | 73,0±1,4 | **0,001** |
| Men (%) | 73.7 |  |  |  |
| Hypertension (%) | 65,3 | 65,3 | 65,4 | 0,986 |
| Obesity (%) | 12,4 | 12,0 | 13,5 | 0,703 |
| Diabetes (%) | 22,8 | 23,4 | 21,2 | 0,644 |
| COPD (%) | 10.4 | 9,6 | 12,5 | 0.409 |
| CKD (%) | 8.6 | 9,3 | 6,7 | 0,427 |
| Coronary artery disease (%) | 15,7 | 16,8 | 12,5 | 0.297 |
| Heart Failure (%) | 14,4 | 12,4 | 23,1 | **0,001** |
| Thyroid Disease (%) | 0,0 | 0 | 0 | 1 |

COPD: Chronic obstructive pulmonary disease; CKD: Chronic kidney disease; p-value for continuous variable Age was calculated by unpaired *t*-test; for categorical variables, the $\chi^2$ test was used.

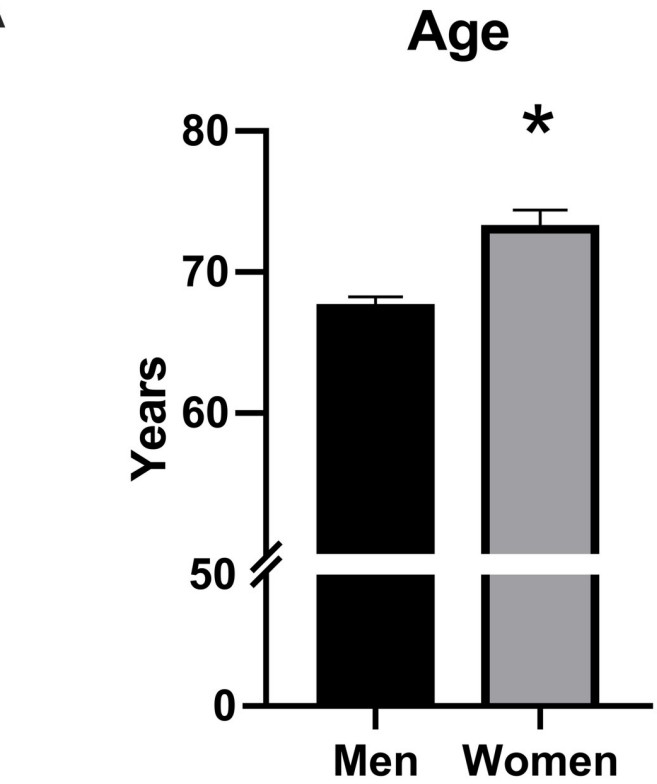

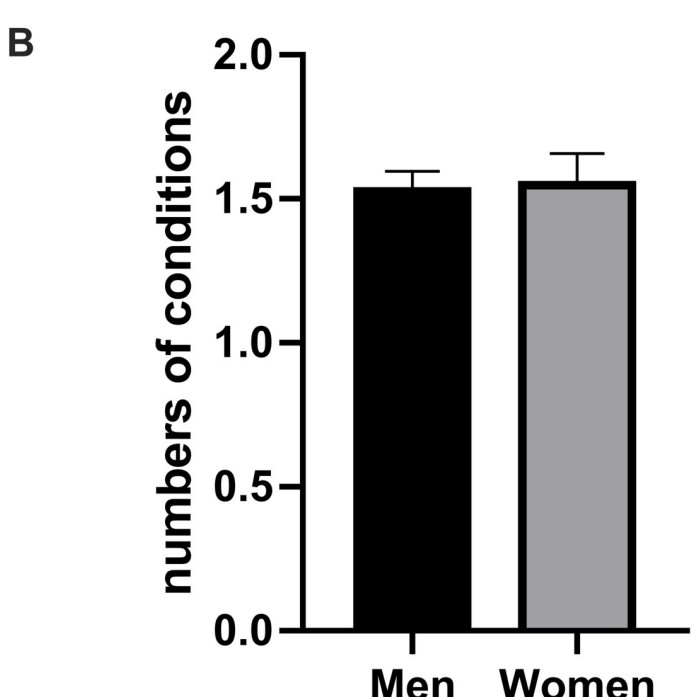

**Fig 1. Panel A** Patients admitted to ICU were 395: Women (n = 104) were older than men (n = 291); * p<0,05, unpaired t test. **Panel B** Men and women admitted at the ICU presented a similar number of comorbidities (hypertension, coronary artery disease, heart failure, diabetes, chronic kidney disease, chronic obstructive pulmonary disease, obesity, history of blood and solid tumors, liver disease).

Clinical characteristics and outcomes of COVID-19 patients differ by gender: males are more prevalent, especially among those admitted to ICU [19] and associated with increased disease severity [20]. A recent study from Meng et al. identified gender-specific differences in disease incidence and fatality rate, related to more severe kidney and liver function abnormalities in male patients [21]. A recent meta-analysis, including more than 59.000 patients identified male gender and older Age as the main predictors of mortality [22]. All these findings, in line with our results, strongly suggest that a gender-specific susceptibility to the infection and disease progression exists. Women—even in postmenopausal Age—seem to be protected against COVID-19 as well as against worse outcomes. A biological mechanism may explain the higher risk of men. The ability of androgens to regulate transmembrane protease serine 2 [23], i.e. the enzyme allowing the final interaction between ACE2 and SARS-CoV-2 and viral RNA entry into the cell, might take part in this mechanism [24]. Also, data from animal studies with SARS-CoV infection have identified more enhanced virus replication and alveolar damage in male mice, mainly due to enhanced and ineffective cytokine response [25].

Recent reports from different countries have highlighted the importance of obesity as primary comorbidity in COVID19 patients leading to ICU admission and deaths [26]. During H1N1 influenza in 2010, different reports identified obesity as the main risk factors for hospitalization and deaths [27]. Obesity is associated with reduced respiratory function due to decreased expiratory reserve, functional capacity, and respiratory system compliance [28]. Furthermore, adipose tissue is involved in complex interactions with the immune system. Release of inflammatory adipokines from visceral fat depots can affect the immune response and contribute to the imbalance between anti and pro-inflammatory adipokines secretion from thoracic visceral fat depots [29]. There is also evidence that obesity impaired adaptive immune response to seasonal influenza virus [30]. This complex scenario can partially explain the cytokine storm described in patients with severe SARS-CoV infection.

The coexistence of male gender and obesity may trigger a maladaptive, exaggerated immune response, leading to the development of acute respiratory distress syndrome, and an increased frequency of ICU admission. Also, among women, admission to ICU was associated

**Table 4. Multivariable logistic regression analysis for ICU admission in the total study population.**

|  | Univariate Analysis | | Multivariable Analysis | | |
|---|---|---|---|---|---|
|  |  |  | Model | | 95% CI |
|  | p | Pearson | p | Beta |  |
| **Age** | 0.425 | 0.016 |  |  |  |
| **Gender (M/F)** | *0.001* | -0.102 | *0.0001* | 1.876 | 1.415–2.304 |
| **Hypertension (y/n)** | *0.003* | 0,061 | *0.023* | 1.314 | 1.039–1.662 |
| **Diabetes (y/n)** | *0.01* | 0,053 | 0.493 | 1.174 | 0.891–1.547 |
| **CKD (y/n)** | *0.005* | 0,058 | *0.034* | 1.588 | 1.036–2.434 |
| **Heart failure (y/n)** | *0.05* | 0,040 | 0.554 | 1.117 | 0.804–1.551 |
| **CAD (y/n)** | 0.400 | 0.017 |  |  |  |
| **Obesity (y/n)** | *0.001* | 0,103 | *0.0005* | 2.476 | 1.724–3,555 |

ICU: intensive care unit; CKD: Chronic kidney disease; CAD: Coronary Artery Disease

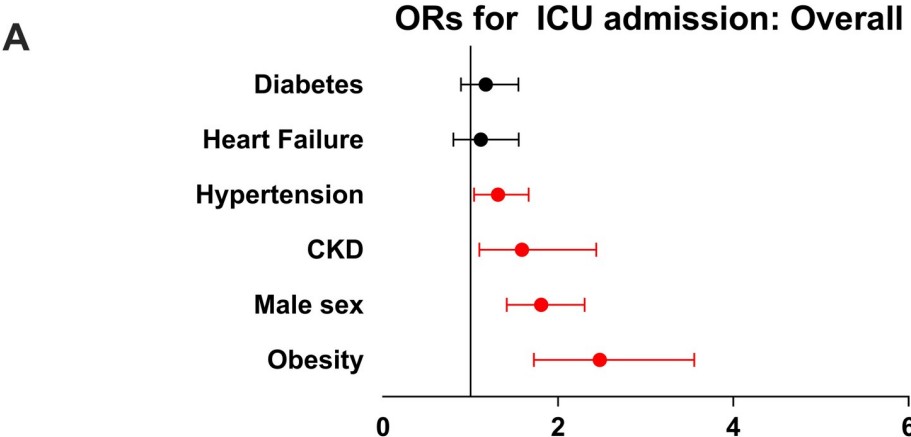

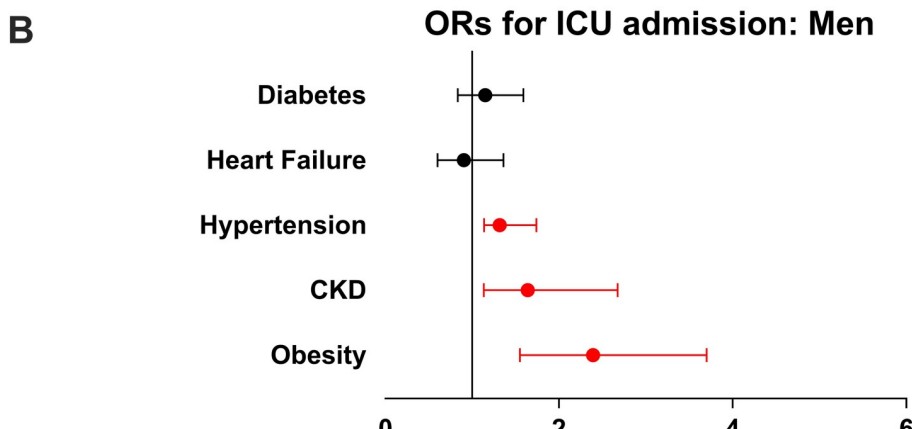

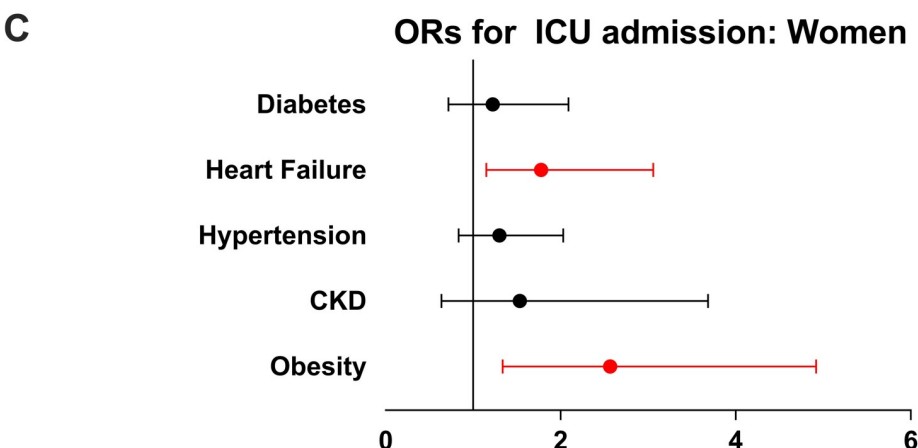

**Fig 2. Panel A:** Forrest plot of the odds ratio and confidence intervals calculated for each variable found to be independently associated with ICU admission in the total study population. **Panel B:** Forrest plot of the odds ratio and confidence intervals calculated for each variable found to be independently associated with ICU admission among

male patients. **Panel C** Forrest plot of the odds ratio and confidence intervals calculated for each variable found to be independently associated with ICU admission among females. **ICU**: Intensive Care Units; In red, p<0,05.

with obesity, supporting the role of decreased inflammatory protection caused by the accumulation of adipose tissue and by increasing Age. With older Age, females develop stronger chronic immune reactions in the myocardium [31]. Heart failure in women, by raising left ventricle filling pressures, could cause a higher frequency of pulmonary complications and hypoxic respiratory failure requiring mechanical ventilation in ICU. Also, sex hormones may affect many components of the renin-angiotensin-aldosterone system, including ACE2. A very recent report shows that women with heart failure have lower plasma ACE2 levels than men [32], although the evidence of the relationship among circulating and tissue concentrations of ACE2 is still missing. In our population, women received fewer ACE inhibitors and more diuretics than men. ACE-inhibitors might exert protection by reducing the production of Angiotensin II, which in turn sustains the pro-inflammatory response [33, 34]. It is possible, therefore, to speculate that reduced use of ACE inhibitors in women also provides reduced protection from COVID-19; at the same time, greater exposure to diuretics could increase the risk of hypokalemia.

Our study has limitations. Firstly, the use of questionnaires might allow for some level of uncertainty. The online questionnaire was designed with closed answers, all necessary for final validation of the form, therefore reducing the risk of incomplete answers. Secondly, the present study includes only symptomatic, hospitalized patients. We excluded subjects with not confirmed disease by nasal or pharyngeal swabs. Therefore, we might have missed those patients with early infection and negative tests. Also, the cross-sectional and not prospective design of our study does not allow us to identify a causative role for the reported parameters on the outcomes.

Nevertheless, our study is compelling in generating hypotheses to be further demonstrated in extensive prospective studies.

## Conclusions

Our study demonstrates a gender effect for women in COVID-19 that are protected from more severe clinical presentations of the disease. In women, heart failure and obesity are more strictly associated with ICU admission.

## Acknowledgments

**The SARS-RAS Investigator group is composed of**:

Arrigo F.G. Cicero 1, Claudia Agabiti Rosei 14, Carlo Aggiusti 14, Fabio Bertacchini 14, Michele Bevilacqua 10, Valeria Bisogni 9, Michele Bombelli 22, Luca Bulfone 16, Flaminia Canichella 18, Giovanni Carpani 22, Massimo Catanuso 17, Carmine Savoia 2, Giulia Chiarini 15, Fernando Chiumiento 21, Giuseppe Mulè 4, Rosario Cianci 6, Giuliano Tocci 2, Franco Cipollini 23, Andrea Dalbeni 10, Roberto Alberto De Blasi 2, Carolina De Ciuceis 14, Raffaella Dell'Oro 22, Antonino Di Guardo 17, Santo Di Lorenzo 2, Monica Di Norcia 25, Roberto Ervo 11, Elisabetta Eula 20, Davide Fabbricatore 14, Elvira Fanelli 20, Claudio Letizia 6, Cristiano Fava 10, Enzo Grasso 12, Alessandro Grimaldi 25, Maddalena Illario 3 (Lead Author for this group; illario@unina.it), Claudio Invernizzi 22, Elena Iraca 1, Federica Liegi 1, Francesca Magalini 26, Paolo Malerba 15, Alessandro Maloberti 12, Martina Mezzadri 6, Giulia Molinari 22, Roberta Mussinelli 5, Anna Paini 14, Paola Pellimassi 2, Paolo Mulatero 20, Ornella Piazza 8, Luigi Pietramala 6, Roberto Pontremoli 24, Fosca Quarti Tevano 22, Franco Rabbia 20,

Monica Rocco 2, Anna Sabena 5, Francesco Salinaro 5, Paola Schiavi 19, Maria Chiara Sgariglia 6, Francesco Spannella 19, Sara Tedeschi 1, Pierluigi Viale 1 and the COVID19 Niguarda group.

The SARS-RAS centers are the following:
1 AO Policlinico Sant'Orsola-Malpighi, Bologna
2 AOU Sant'Andrea, Roma
3 AOU Federico II, Napoli
4 AOU Policlinico Paolo Giaccone, Palermo
5 AOU Policlinico San Matteo, Pavia
6 AOU Policlinico Umberto I, Roma
7 AOU Policlinico Universitario, Padova
8 AOU San Giovanni di Dio e Ruggi d'Aragona, PO "Dell'Olmo" Cava de' Tirreni
9 AOU Santa Maria, Terni
10 AOUI Verona, Italy
11 ASL 1 Imperiese, Ventimiglia
12 ASST Grande Ospedale Metropolitano Niguarda, Milano
13 ASST Santi Paolo e Carlo, Milano
14 University of Brescia & ASST SPEDALI CIVILI BRESCIA
15 University of Brescia & ASST SPEDALI CIVILI. PO Montichiari
16 ASUI Friuli Centrale, Udine
17 Centro Ipertensione Mascalucia, Catania
18 INMI Lazzaro Spallanzani, Roma
19 INRCA, Ancona, Italy
20 Ospedale "Le Molinette", Torino
21 Ospedale di Eboli, Salerno
22 Ospedale San Gerardo, Monza
23 Ospedale San Jacopo, Pistoia
24 Ospedale San Martino, Genova
25 PO San Salvatore, L'Aquila
26 Ospedale Maggiore, Parma

The authors dedicate this work to the memory of Maurizio Galderisi, MD, a respected scientist, a man of great humanity, a friend to all of us at the Italian Society of Hypertension, a great loss caused by this terrible disease.

## Author Contributions

**Conceptualization:** Guido Grassi, Claudio Ferri.

**Data curation:** Guido Grassi, Stefano Carugo, Davide Grassi, Pietro Minuz, Massimo Salvetti.

**Formal analysis:** Guido Grassi, Claudio Ferri.

**Investigation:** Claudio Borghi, Stefano Carugo, Francesco Fallo, Cristina Giannattasio, Davide Grassi, Claudio Letizia, Costantino Mancusi, Pietro Minuz, Stefano Perlini, Giacomo Pucci, Damiano Rizzoni, Riccardo Sarzani, Leonardo Sechi, Franco Veglio.

**Methodology:** Francesco Fallo, Cristina Giannattasio, Costantino Mancusi, Stefano Perlini, Giacomo Pucci, Damiano Rizzoni, Riccardo Sarzani, Franco Veglio.

**Project administration:** Claudio Borghi.

**Resources:** Claudio Borghi.

**Supervision:** Guido Iaccarino, Massimo Volpe.

**Validation:** Guido Iaccarino, Leonardo Sechi, Massimo Volpe.

**Writing – original draft:** Guido Iaccarino, Costantino Mancusi, Maria Lorenza Muiesan.

**Writing – review & editing:** Guido Iaccarino, Maria Lorenza Muiesan.

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
