## [Decision Letter · Decision Letter 0]

22 Jun 2020

PONE-D-20-14687

GENDER DIFFERENCES IN PREDICTORS OF INTENSIVE CARE UNITS ADMISSION AMONG  COVID-19 PATIENTS:The results of the SARS-RAS study of the Italian Society of Hypertension

PLOS ONE

Dear Dr. Iaccarino,

Thank you for submitting your manuscript to PLOS ONE. After careful consideration, we feel that it has merit but does not fully meet PLOS ONE’s publication criteria as it currently stands. Therefore, we invite you to submit a revised version of the manuscript that addresses the points raised during the review process.

Kind regards,

Tatsuo Shimosawa, M.D., Ph.D.

Senior Academic Editor

PLOS ONE

---

## [Author Response · Author response to Decision Letter 0]

6 Jul 2020

Reviewer #1.

Reviewer #1: The study "GENDER DIFFERENCES IN PREDICTORS OF INTENSIVE CARE UNITS ADMISSION AMONG COVID-19 PATIENTS: The results of the SARS-RAS study of the Italian Society of Hypertension" by Guido Iaccarino, et al. showed the importance of sex difference as the main determinant of the disease’s severity among COVID-19 patients. The study is interesting and well written. My comments are below.

We thank this Reviewer for the favorable comment finding our paper “interesting and well written”.

Major comments

#1. The predictors of ICU admission have been centrally discussed, and its relation to sex differences is an important and interesting point. However, most of the discussions have been limited to the general risk factors for ICU admission in COVID-19 patients, such as sex or obesity, and there is little mentioned about the difference in risk factors by sex. The authors should discuss more deeply about this issue with objective data presentation, including multiple analyses of other previous studies, as well as reference to the results of basic research as necessary.

We agree with this comment and have expanded the characterization of the phenotypes of patients admitted to ICU admission according to gender. It is now clear from the new table that women admitted to ICU are older and with increased frequency of HF diagnosis , but not of CAD when compared to men. This is in agreement with the epidemiology of these conditions, for which at older ages more females than males are affected by HFpEF. We have added a new table 3, which is presented in the results at lines 1-3, page 9 and discussed at lines 17-25 of page 10. 

#2. There is serious drawback of the study due to nature of data collection through questionnaire. It is not clear whether comorbidities, including diabetes and hypertension, were properly controlled. Moreover, the rationale for defining a BMI≥ 30 as obese should be given (BMI ≥ 25 or BMI ≥ 35 as the predictor for ICU admission is also interesting). This information is inevitable to discuss obesity or hypertension as the risk factor for ICU admission.

We agree that the use of a questionnaire represents a limitation of the study, however many of the data collected were verifiable. For example, we verified that the presence of a positive history for hypertension or coronary disease corresponded to the prescription of cardiovascular drugs. Similarly, heart failure and diabetes were also controlled. We used the BMI >30 as a cutoff, according to the classification of the Center of Disease Control. We did not collect weight and height, and BMI were calculated at the hospital premises. Obesity and hypertension were independently analyzed in univariate and multivariable analysis. 

#3. Mortality data and length of ICU stay should be evaluated.

We do not have data regarding the length of stay in ICU, given the cross-sectional nature of our survey. Regarding mortality, the determinants of death in our population was the objective of a recently published paper, which, by the way, shows that sex is not relevant to this outcome (doi/10.1161/HYPERTENSIONAHA.120.15324)

Minor comments

#1. The quality of the figure can be improved.

We have increased the quality to 600 DPI

#2. In Table2, the number of patients who admitted to ICU or not is wrong?

We have updated the correct numbers of Table 2, thanks for pointing it out

Reviewer #2

1. The manuscript is technically sound and data support the conclusion

We thank this Reviewer for the favorable comment.

2. The statistical analysis has been performed appropriately. However I want to make the following comments:

a. The authors do nowhere describe the level of data completeness. Out of the 2378 questionnaires (NB Table 1 states 2377!) that were collected was each and every one of these questionnaires completely filled out with unambiguous information of all variables asked for? If no, how was missing data treated in the statistical analyses?

We now better indicate that indeed, the number of questionnaires collected was larger than that used for the statistical analysis. We collected 2409 questionnaires, of which 2378 were used for analysis, as they contained all data. The questionnaire had closed answers, to be chosen between a limited number of responses, mostly only 2. We verified that the data were complete, entered correctly and that there were no duplications.

b. Since authors refer to validation-procedures (p 10, row 6) with laboratory findings at diagnostic laboratories at each participating institution there is presumably a possibility to check and account for how the coverage (and coverage in males vs females) of covid-19 sick hospitalized patients the questionnaire survey accounted for.

We had agreement with individual wards or departments, therefore we do not have the whole hospital number of admissions for COVID-19. Therefore, we cannot point out the coverage of covid-19 patients that the hospital accounted for. We have for sure, full coverage of the COVID-19 patients admitted with the indicated departments in the time frame indicated in the manuscript.

c. It is somehow unclear how the author reasoned when including and excluding potential explanatory variables in the multivariate logistic analysis presented in Table 3. This is mentioned briefly, but not to satisfactory extent, on page 10, row 16-17. When reviewing the observational demographics statics in Table 1 it appears as if there are variables that to me are both “significant and clinically relevant” to include in the modelling. For example, why include heart failure (where there seem to be no difference between groups in Table 1) but not include coronary artery disease where there appear to be a clear gender difference? One cannot help asking one selves if the “significant beta” for gender just reflects a “significant beta” for coronary artery disease between men and women admitted to the ICU? 

We calculated the Pearson correlation of each variable (age, sex, CAD, HF, CKD, COPD, hypertension, Diabetes, obesity) and ICU admission. Afterwords, we included in the multivariable analysis those variables that had a statistical significant Pearson correlation with the ICU admission outcome. We now show these parameters in two columns added in the new table 4.

d. Was data completeness and questionaries design of such quality that it was possible to collect and present reliable Charlson Co-morbidity index? I could only access one of two supplementary questionnaires and these were written in Italian. If the index was complete and of high quality for most subjects – why was it not included as a potential explanatory variable in the multivariate logistic regression analysis? If data on the index was poor – the authors should consider omitting it.

We take advantage of this Reviewer suggestion and remove data regarding the Charlson index, which is the objective of a recently published paper (https://www.ahajournals.org/doi/10.1161/HYPERTENSIONAHA.120.15324)

e. In Figure 1, panel B, the authors referrer to “number of co-morbidities” in males and females admitted to the ICU. However it is not clear exactly how many possible potential co-morbiditites that actually were includes and accounted for in Figure 1, panel B. Is it “only” the conditions presented as potential explanatory variables in Table 3 (i.e. Hypertension, Diabetes, CKD, Heart failure and Obesity) or is the vast range of comorbidities that are presented in the method sections (page 9, row 21 to page 10, row 3).

Thanks for pointing this out. We have now clarified in the text that the means reflect the whole number of comorbidities listed in the methods.

f. I do not really understand the rational of including observational cardiovascular medication treatment data (Table 2). To present this kind of data is not stated as an objective of the paper. The results in this table are also not commented in the results section. There is however a section in the Discussion (page 13, row 17) about ACE-inhibitors but the language in this section is particularly challenging to understand.

We have now added a comment in the Background (lines 22-24 of page 5) regarding the use of ACE inhibitors and AT1 antagonists as a putative mechanism of COVID-19, and the aim to discriminate a sex dependent effect of this class of drugs. We have also reworded the related Discussion to make it clearer.

3. See comments under bullet #2.

See reply to bullet #2

4. The language and wording in the manuscript is at times not eloquent and challenging to understand. I advise the authors to consult a language-editor.

We checked the manuscript with an English mother-tongue editor.

5. In the methods section (p 10, row 8-9) the authors state that they did collect outcome (alive hospital dismission or in-hospital mortality) data for the studied population. If this data was collected and is available - why not present this data in the paper to give the reader an idea of whether there are gender differences in the population admitted to the ICU and those not admitted to the ICU with respect to outcome?

We thank the Reviewer for pointing out this important aspect. ICU admission increases the risk of death in the overall population, with an OR[IC] 4,189[3,184-5,511]. Among patients admitted to the ICU, the majority were men. The OR[IC] of death among ICU patients was affected by the female gender (1,784[1,103-2,885]), older age (1,094[1,068-1,120]), and number of comorbidities (2,243[1,667-3,017]). When we ran the multivariable analysis, comorbidities (1,801[1,310-2,476]) and age (1,082[1,044-1,121]) remained significant modifiers of the OR for death, while female sex was no longer statistically significant (1,535[0,657-3,586]).

We propose this subanalysis to this Reviewer for perusal, nevertheless, we rather prefer not to include it in the present manuscript, since a more extensive analysis of the hard outcome death is the objective of a recently published paper (doi/10.1161/HYPERTENSIONAHA.120.15324).

---

## [Decision Letter · Decision Letter 1]

21 Jul 2020

PONE-D-20-14687R1

GENDER DIFFERENCES IN PREDICTORS OF INTENSIVE CARE UNITS ADMISSION AMONG  COVID-19 PATIENTS:The results of the SARS-RAS study of the Italian Society of Hypertension

PLOS ONE

Dear Dr. Iaccarino,

Thank you for submitting your manuscript to PLOS ONE. After careful consideration, we feel that it has merit but does not fully meet PLOS ONE’s publication criteria as it currently stands. Therefore, we invite you to submit a revised version of the manuscript that addresses the points raised during the review process.

We look forward to receiving your revised manuscript.

Kind regards,

Tatsuo Shimosawa, M.D., Ph.D.

Academic Editor

PLOS ONE

Reviewers' comments:

Reviewer's Responses to Questions

**Comments to the Author**

1. If the authors have adequately addressed your comments raised in a previous round of review and you feel that this manuscript is now acceptable for publication, you may indicate that here to bypass the “Comments to the Author” section, enter your conflict of interest statement in the “Confidential to Editor” section, and submit your "Accept" recommendation.

Reviewer #1: All comments have been addressed

Reviewer #2: All comments have been addressed

2. Is the manuscript technically sound, and do the data support the conclusions?

Reviewer #1: Yes

Reviewer #2: Yes

3. Has the statistical analysis been performed appropriately and rigorously? 

Reviewer #1: Yes

Reviewer #2: Yes

4. Have the authors made all data underlying the findings in their manuscript fully available?

Reviewer #1: Yes

Reviewer #2: Yes

5. Is the manuscript presented in an intelligible fashion and written in standard English?

Reviewer #1: Yes

Reviewer #2: No

6. Review Comments to the Author

Reviewer #1: The manuscript has been much improved and is in a nice condition now.

The authors have answered my concerns as much as possible.

Reviewer #2: Thank you for the revised version of the manuscript where I also enjoyed to learn about the input from my reviewer-colleague. Most of my input has been accounted for in the revised version. Thank you indeed. The current manuscript-version is of better quality. However, despite the authors claiming that a native-English speaker is responsible of the wording, a range of linguistic, syntax and grammar errors still occur in the manuscript. It is readable, but it is sometimes hardly intelligible. I therefore still recommend an additional language review but leave the final decision about this to the PLOS one editor since the standard of the accepted paper reflects upon the journal. I do not need to see the paper again but am happy to reccommend that it is accepted for pubilication.

Below I list some EXAMPLES that I encountered in the text, just to illustrate that there are room for language and editiing-improvements:

P3.

R19: exchange “on the opposite, among women” to “whereas in women”

R21: omit “significantly”. In an abstract only significant findings should be described.

R23: omit “biological”. Authors have not provided evidence enough to state that there are biological factors behind the observed results.

Sometimes reference-brackets appear after the punctuation symbol “.” and sometimes before it. Overall there are several small errors and omitted spaces in association with reference-brackets.

P5

R18: exchange “smoke” with “smokers”. Or re-phrase the sentence completely.

P6

R3: Consider omitting “an”

R24: exchange “one or more physicians” with “at least one physician”

P8

R15: Decide and be consistent of whether to write “Ace” och “ACE”. You are not consistent now.

R20: exchange “or” at the end of the row with “and patients”

P9

R1 For consistency exchange “female patients” with “females” since that is the formed used for “males”

R8 Same comment as for abstract. Why using “significantly” here?

P11

R16: A punctuation is lacking.

R20: poor spacing

R21/22. Poor wording of sentence.

R22/23 Poor wording of sentence.

R25 Wrong use of capitalisation

P12

R1: Exchange “some limitations” to “limitations” Exchange “First” to “Firstly”

R5: Exchange “Second” to “Secondly”.

P13

R:21: omit the first “of” and insert a “the” in front of “Italian..”

7. PLOS authors have the option to publish the peer review history of their article (what does this mean?). If published, this will include your full peer review and any attached files.

Reviewer #1: No

Reviewer #2: No

---

## [Author Response · Author response to Decision Letter 1]

22 Jul 2020

Reviewer #1.

Thank you.

Reviewer #2

Thank you for your revision. We have included the suggested changes and reworded throughout the manuscript the less fluent parts.

---

## [Editor Report · Decision Letter 2]

24 Jul 2020

GENDER DIFFERENCES IN PREDICTORS OF INTENSIVE CARE UNITS ADMISSION AMONG  COVID-19 PATIENTS:The results of the SARS-RAS study of the Italian Society of Hypertension

PONE-D-20-14687R2

Dear Dr. Iaccarino,

We’re pleased to inform you that your manuscript has been judged scientifically suitable for publication and will be formally accepted for publication once it meets all outstanding technical requirements.

Kind regards,

Tatsuo Shimosawa, M.D., Ph.D.

Academic Editor

PLOS ONE
---

## [Editor Report · Acceptance letter]

28 Jul 2020

PONE-D-20-14687R2 

GENDER DIFFERENCES IN PREDICTORS OF INTENSIVE CARE UNITS ADMISSION AMONG  COVID-19 PATIENTS:The results of the SARS-RAS study of the Italian Society of Hypertension 

Dear Dr. Iaccarino:

I'm pleased to inform you that your manuscript has been deemed suitable for publication in PLOS ONE. Congratulations! Your manuscript is now with our production department. 

Kind regards, 

on behalf of

Prof. Tatsuo Shimosawa 

Academic Editor

PLOS ONE